# Numerical Simulation of Microscale Oblique Droplet Impact on Liquid Film

Yan Cao [1], Jingxin Wang [1] and Chunling Zhu [1,2,*]

1    College of Aerospace Engineering, Nanjing University of Aeronautics and Astronautics,
     Nanjing 210016, China
2    State Key Laboratory of Mechanics and Control of Mechanical Structures, Nanjing University of Aeronautics
     & Astronautics, Nanjing 210016, China
*    Correspondence: clzhu@nuaa.edu.cn

**Abstract:** The oblique impact of microscale water droplets on liquid film is numerically investigated. Two-phase flow problems are simulated using three-dimensional incompressible Navier-Stokes equations, and the level-set method is employed for capturing the gas-liquid interface. The numerical model is verified using experimental results from a normal and oblique impact via the qualitative comparison of crown profile features and quantitative contrast of the crown height and radius varying with time. The article discusses the influence of tangential impact velocity, water film thickness, Reynolds number, and Weber number on the shape characteristics, tangential momentum, and kinetic energy of the annular crown. The results show that the decreasing momentum in the tangential direction can be divided into three clear stages: rapid decrease, slight increase, and continuous decrease. In addition, film thickness and Weber number have significant effects on the momentum decay rate.

**Keywords:** level-set method; oblique impact; liquid film; momentum decay

## 1. Introduction

Gas-liquid interface evolution is a unique physical phenomenon in which droplets impact on a liquid film, and it has been extensively studied due to its natural and industrial applications, such as spray cooling [1], inkjet printing [2], and droplet impact on the surface of an aircraft causing aircraft icing. In general, droplet impact on a dry surface occurs only at the initial stage of impinging; then the surface is quickly wetted by high-speed droplets and covered by a liquid film. In addition, most droplets obliquely impact a liquid film at a specific angle. Only a few droplets vertically impact the target surface. Therefore, it is of great practical significance to study the mechanism underlying oblique liquid droplet impact on liquid film.

Several early works [3–6] have focused on the dynamics of droplet impact on liquid film, including bouncing, coalescence, and splashing. The regime of these phenomena is related to droplet diameter, velocity, film thickness, and the physical parameters of droplets, film, and surrounding gas. From a physical perspective, the different categories of phenomena produced by droplet impact on a liquid film are derived from the interactions among inertial force, viscous force, and surface tension, which can be characterized by dimensionless parameters *Re*, *We*, and *Oh*:

$$Re = \frac{\rho_l U_d D_d}{\mu_l}, \ We = \frac{\rho_l U_d^2 D_d}{\sigma}, \ Oh = \frac{\mu_l}{\sqrt{\rho_l \sigma D_d}}, \tag{1}$$

where $\rho_l$, $\mu_l$, and $\sigma$ are the liquid density, viscosity, and surface tension, respectively. $D_d$ and $U_d$ are the droplet diameter and impact velocity, respectively. In addition, the

nondimensional film thickness is also an important factor on droplets impact, which is defined as:

$$H = \frac{h}{D_{\mathrm{d}}},$$

(2)

where $h$ is the thickness of the water film.

Many experiments and numerical simulations have been carried out to study the crown splashing phenomena of a normal impact on a liquid film. Krechetnikov and Homsy [7] found that the Richtmyer-Meshkov instability mechanism plays a dominant role at short times through kinematic measurements. Gao et al. [8] experimentally studied crown splash evolution over time. Using previous studies, the model for predicting crown evolution is modified by researchers and extended to the problem of impact on a flowing liquid film. Weiss and Yarin [9] found that the crown splashing threshold is a function of the impact parameters by experimentally and theoretically examining the continuous impact of droplets. Mukherjee et al. [10] employed the lattice Boltzmann model to simulate the axisymmetric crown splashing of a droplet impact. The radius and height of the crown increased with the increasing film thickness when the droplets impacted on thin ($H < 0.25$) film, but decreased when the thick liquid film ($H > 0.25$) was impacted. A three-dimensional numerical simulation study of droplet impact on liquid film was conducted by Nikolopoulos et al. [11], which used an adaptive local grid refinement technique to simulate secondary droplets accurately. Their result revealed that the appearance of secondary droplets is caused by Rayleigh instability in the early stage of impact, while in the later stage, the surface tension dominates. Lee et al. [12] employed the level-set method to study the splash phenomenon in two-dimensional axisymmetric coordinates. Cheng et al. [13] introduced an artificial perturbation shortly after the moment of impact to simulate crown spike formation by using a two-phase flow lattice Boltzmann model. In addition, results by Cheng for liquid droplets impinging on a moving-wall liquid film showed that different moving velocities of the wall could enhance or inhibit crown evolution. The numerical research by Shetabivash et al. [14] indicated that an increase in gas density reduced the evolution speed of the crown radius and height, and that the Reynolds number determined the crown height. Moreover, they proposed that the crown radius of prolate droplets would develop faster than the crown radii of spherical and flat droplets. Rahmati et al. [15] found that on hydrophobic surfaces, the splash from a droplet impact on liquid film was restrained. They also simulated the vertical impact of two droplets arranged horizontally and then vertically on a stationary or moving liquid film. The impact of a liquid drop onto a thin liquid film of different fluid is investigated experimentally by Hannah et al. [16]. According to their theoretical analysis, the splash is determined by the critical number $K_d$ (based on the properties of a droplet) or $K_f$ (based on the properties of film) for different viscosity ratios of film and droplet.

Due to the tangential velocity, the flow characteristics of oblique drop impact on thin liquid film are asymmetric in the tangential direction. To examine the differences between oblique and normal drop impact, several experimental and numerical simulations have been conducted. Okawa et al. [17] used a high-speed camera to observe a ship's prow-like asymmetric liquid sheet and secondary water droplets downstream caused by oblique impact. Their results showed that the impingement angle significantly influences the drop impact. Che et al. [18] observed oblique drop impact on a falling liquid film and divided the observed phenomena into rebound, partial coalescence, total coalescence, and splashing classes. Gielen et al. [19] studied the oblique impact of 100 μm droplets on a deep pool and classified the phenomena after impact: deposition, partial splash, and full splash. Scaling arguments that delineated these regimes using the Weber number and impact angle were provided. Cheng et al. [20] discussed drop impact on static and flowing liquid films, with an impact angle of 0~60°. They used the two-phase lattice Boltzmann model, and a critical angle was proposed beyond which the splash of the downstream crown was completely suppressed. Guo and Lian [21] employed the moment-of-fluid (MOF) method to simulate a high-speed oblique drop impact on liquid film. The results

showed that the tangential velocity, the main factor affecting the splash downstream of the impact, significantly affected the lamella height and radius, as well as the vortices at the drop-film interface. Chen et al. [22] used the Lattice Boltzmann method to simulate the splash dynamics mechanism of three-dimensional droplet oblique impact on liquid film. The effects of different factors on crown evolution were compared in detail, and a correlation of splash limit with Ohnesorge number and the Weber number was established at a medium impact angle. Wang et al. [23] obtained the heat transfer characteristics of a drop impact using the coupled level-set and volume if fluid (CLSVOF) method. The impact angle affected the local surface heat flux distribution at the impact location. However, the average heat flux of the whole surface was not affected by a change in impact angle. Liu et al. [24] used the lattice Boltzmann method to simulate oblique droplet impact on film for a given velocity and gravity, and found that similar to the normal impact, the oblique impact also exhibited a critical film thickness. When the film thickness is lower than this critical thickness, the crown height downstream increased with the increasing water film thickness, while if the thickness is higher than the critical thickness, the trend is opposite. Guo et al. [25] also obtained this rule by adopting the CLSVOF method, and their simulation results revealed that the critical water film thickness was greater at a low Weber number. Bao et al. [26] conducted a numerical study on the dynamic characteristics of a continuous oblique impact of two droplets on thin liquid film using the CLSVOF method.

The above research has elucidated the crown evolution and splashing of normal and oblique drop impact on liquid film. However, the changes in momentum and kinetic energy of the droplet and film during impact have not yet been explored in detail, and the relationship among momentum as well as kinetic energy and Re, We, and H are not clearly illustrated, which is crucial in the aircraft icing problem where the flow of the water film is considered on a macroscopic level. In the present study, we numerically investigate the oblique drop impact on a static liquid film with the droplet diameter range of 10–200 μm and the impact velocity range of 30–60 m/s, focusing on the geometric features, change in momentum, and dissipation of kinetic energy during impact. The investigation is conducted by considering different impact angles, film thicknesses, Reynolds numbers, and Weber numbers.

## 2. Materials and Methods

### 2.1. Level-Set Method

The level-set method [27,28] is applied in this study for tracking the two-phase interface. The level-set function $\phi$ is a signed distance function based on the interface $\Gamma$, and the sign represents different areas divided by the interface $\Gamma$. The function $\phi$ is defined as:

$$\phi(x) = \begin{cases} \min\|x - x_\Gamma\|, & x \in \text{ the gas} \\ 0, & x \in \Gamma \\ -\min\|x - x_\Gamma\|, & x \in \text{ the liquid.} \end{cases} \tag{3}$$

The evolution of the interface is tracked by solving the level-set transport equation:

$$\frac{\partial \phi}{\partial t} + \boldsymbol{u} \cdot \nabla \phi = 0. \tag{4}$$

A reinitialization equation is introduced to maintain the level-set function $\phi$ as a signed distance function after solving the transport equation in each time step:

$$\frac{\partial \phi}{\partial \tau} + Sign(\phi_0)(|\nabla \phi| - 1) = 0, \tag{5}$$

where $\phi_0$ is the signed distance function $\phi$ before reinitialization, $Sign(\phi_0)$ is the sign function, and $\tau$ is the pseudo marching time. In the present study, $\tau = 0.3\Delta x$. In general,

after solving the level-set transport equation, it is only necessary to solve the reinitialized equation a few times to maintain $\phi$ as the signed distance function.

In this study, a level-set re-distancing [29] method is used to reduce the interface error. The core concept is that $\phi$ on the grid point closest to the interface is not changed during reinitialization. The reinitialization equation then becomes:

$$\frac{\partial \phi}{\partial \tau} + Sign(\phi_0)(|\nabla \phi| - 1)(1 - \delta_d(\phi_0)) = 0, \tag{6}$$

where $\delta_d(\phi_0)$ is defined as:

$$\delta_d(\phi) = \begin{cases} 1, & \phi(x)\phi(x + \Delta x < 0) \\ 0, & \text{otherwise.} \end{cases} \tag{7}$$

The smoothed Heaviside function $H(\phi)$ is applied for smoothing the discontinuous physical properties near the interface, which is defined as:

$$H(\phi) = \begin{cases} 1, & \phi > \varepsilon \\ \frac{1}{2}\left[1 + \frac{\phi}{\varepsilon} + \frac{1}{\pi}\sin\left(\frac{\pi\phi}{\varepsilon}\right)\right], & |\phi| \le \varepsilon \\ 0, & \phi < -\varepsilon, \end{cases} \tag{8}$$

where $\varepsilon$ represents half of the thickness of the smoothing band between the liquid and gas. In the present study $\varepsilon = 1.5\Delta x$. The distributions of physical properties, such as density and viscosity, are determined by

$$\begin{aligned} \rho(\phi) &= \rho_l + (\rho_g - \rho_l)H(\phi), \\ \mu(\phi) &= \mu_l + (\mu_g - \mu_l)H(\phi), \end{aligned} \tag{9}$$

where the subscripts $l$ and $g$ refer to the liquid and gas phase, respectively.

A third-order Runge–Kutta scheme is applied for the temporal discretization of the transport equation, and a second-order Runge–Kutta scheme is applied for the temporal discretization of the reinitialization equation. A fifth-order WENO scheme is applied to discretize the first derivative of the level-set transport equation and the reinitialization equation.

### 2.2. Governing Equations

The mass and momentum conservation equations of an incompressible, immiscible, multiphase flow are as follows:

$$\nabla \cdot \boldsymbol{u} = 0, \tag{10}$$

$$\frac{\partial \rho \boldsymbol{u}}{\partial t} + \nabla \cdot \rho \boldsymbol{u}\boldsymbol{u} = -\nabla p + \nabla \cdot \left[\mu\left(\nabla \boldsymbol{u} + \nabla \boldsymbol{u}^T\right)\right] + \rho \boldsymbol{g} + \boldsymbol{f}_s, \tag{11}$$

in which $\boldsymbol{u}$ is the velocity vector, $p$ is the pressure, $\boldsymbol{g}$ is the gravitational acceleration, and $\boldsymbol{f}_s$ is the surface tension acting on the gas–liquid interface, which is calculated by the continuum surface tension force model:

$$\boldsymbol{f}_s = \sigma \kappa \nabla \widetilde{C}, \tag{12}$$

where $\sigma$ is the surface tension coefficient, $\widetilde{C}$ is a mollified version of the Heaviside function [30], and $\kappa$ is the interface curvature:

$$\kappa = -\nabla \cdot \boldsymbol{n}, \boldsymbol{n} = \frac{\nabla \phi}{|\nabla \phi|}, \tag{13}$$

$\boldsymbol{n}$ is the unit normal vector computed using the signed distance function $\phi$. In this work, the PISO algorithm [31] is used to couple the pressure with the velocity on the collocated grid, and the momentum interpolation method (MIM) introduced by Rhin-Chow [32] is applied to avoid pressure oscillation. The convection term of the momentum equation is

discretized by a second-order upwind scheme, and the viscous term is discretized by the central difference scheme.

The high-order momentum-preserving method as used by Desmons and Coquerelle [33] is adopted to eliminate numerical errors due to discretization. This method introduces a special continuity equation, which is solved by the same numerical scheme as the advection terms of the momentum equation:

$$\frac{\partial \chi}{\partial t} + \nabla \cdot \boldsymbol{u}\chi = 0, \tag{14}$$

in which $\chi$ is the predicted density $\rho^*$ or viscosity $\mu^*$.

The numerical calculation strategy is:

1. Solve the level-set transport equation (Equation (4)) and the reinitialization equation (Equation (6)) to obtain $\phi^{t+1}$, and update $\rho^{t+1}$, $\mu^{t+1}$;
2. Solve Equation (14) to obtain $\rho^*$, $\mu^*$;
3. Solve the momentum equation to obtain $(\rho\boldsymbol{u})^*$, and then calculate $\boldsymbol{u}^* = \frac{(\rho\boldsymbol{u})^*}{\rho^*}$;
4. Solve the pressure Poisson equation and update $p^{t+1}$;
5. Update the velocity $\boldsymbol{u}^{t+1}$.

## 3. Model Validation

To illustrate the present numerical method qualitatively, three experiments of normal and oblique droplet impact on film were conducted and the experimental parameters are shown in Table 1. The experimental setup used to study a droplet impacting on a thin water film is shown in Figure 1. The study used a smooth aluminum plate (20 mm × 80 mm) with a roughness of 0.05 µm and a static contact angle of 43.8° after being polished by 5000-mesh emery paper. Careful cleaning of the aluminum plate was carried out by using an ultrasonic bath. The plate was hydrophilic to ensure uniformity of the thin water film. Deionized water was used as the droplet and thin film fluid. Film thickness was measured using a chromatic confocal displacing sensor, with a precision of 1 nm (STIL Initial4, CL4MG35, France). The average film thickness was calculated using five values measured near the impact point, as shown in Figure 2. The measurement data of film thickness in Case A is also provided to prove the uniformity of film. A droplet was produced by a syringe pump, with a droplet forming at the tip of a dispensing needle (inner diameter of 0.11 mm). The droplet detached from the needle under gravity and rolled on the homemade water-repellent guideway. The impact velocity and incident angle were adjusted by changing the guideway height and tilt angle. The impact characteristics were captured by a high-speed camera (PCO.DIMAX Germany) with a micro-lens (100 mm F2.8, Tokina). The frame rate of the camera was 5000 fps, and the image resolution was 12 µm per pixel. The droplet diameter was obtained by image analysis [34,35], and the impact velocity was obtained through successive image sequences by using the MATLAB program, similar to literature [34,36]. The visibility of the gas-liquid interface was enhanced by a LED light source, which provides back-lighting and illuminates the droplet to produce a shadowgraph. The ground glass is used to homogenize the light as a diffuser. The experiments were performed at room temperature and atmospheric pressure.

**Table 1.** Experimental parameters of normal and oblique droplet impact on film.

| | Droplet Diameter (mm) | Impact Velocity (m/s) | Film Thickness (µm) | Impact Angle |
|---|---|---|---|---|
| Case A | 2.21 | 3.126 | 207 | normal |
| Case B | 2.21 | 3.126 | 548 | normal |
| Case C | 4.10 | 1.67 | 215 | 30° |

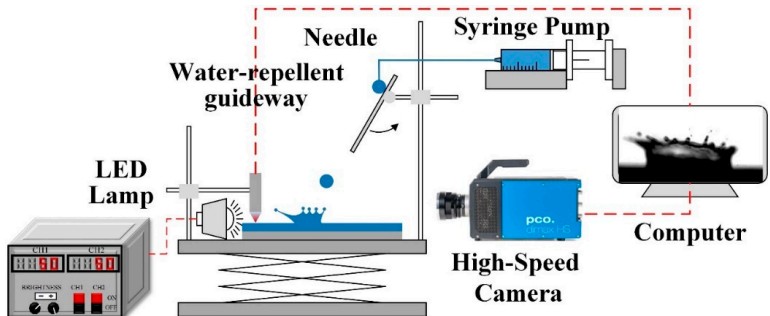

**Figure 1.** Schematic of the experimental setup for droplet impact.

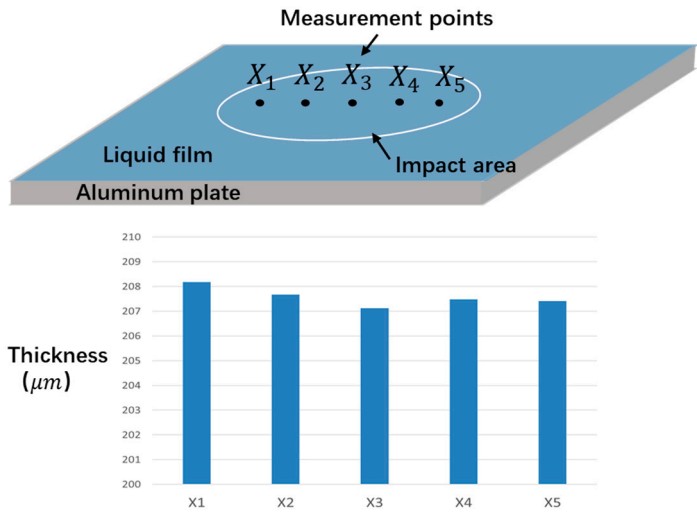

**Figure 2.** Measurement method of film thickness and the data of Case A.

A schematic illustration of a droplet impacting on a liquid film is provided in Figure 3. We first validated the numerical method via two experiments concerning normal droplet impact on a liquid film. In the initial stage of simulation, a liquid water film with a dimensionless height $H = h/D_d$ was set at the bottom surface of the computational domain, and a droplet set at the center of the computational domain above the film, with diameter $D_d$ and velocity $U_d$. The non-dimensional time is described as $T = tU_d/D_d$, where t is the physical time. The time $T = 0$ is defined as the moment when the droplets contact with the film surface. The size of the computational domain was $4D_d \times 4D_d \times 2D_d$, which was verified to be large enough to capture the evolution of crown splashing. The grid resolution was 100 grids per droplet diameter. Only a quarter of a droplet was calculated because of the symmetry of the normal droplet impact. A no-slip boundary condition was applied to the bottom surface, while pressure outlet boundary conditions were adopted for the top and side surfaces.

After that, a three-dimensional, oblique droplet impact on a liquid film was simulated. The impact angle is $\theta$, and $U_{dt}$ and $U_{dn}$ are the tangential and normal components of the impact velocity $U_d$, respectively. The initial droplet center was set at $(X_d, Y_d, Z_d) = (-1.0, 0.0, 0.0)$, and the physical parameters were the same as those of the numerical simulations of normal drop impact. The size of the computational domain was $5D_d \times 4D_d \times 1.5D_d$, which was proved to be large enough to capture the interface evolution. The grid resolution was 100 grids per droplet diameter. Only half of the droplet was calculated because of the bilateral symmetry of the oblique droplet impact. A no-slip boundary condition was applied for the bottom surface, while pressure outlet boundary conditions were adopted for the top and side surfaces.

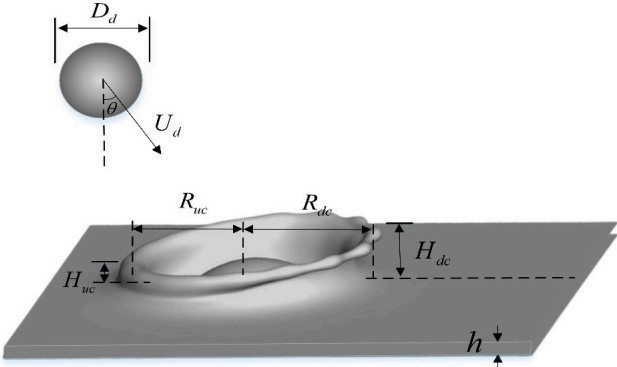

**Figure 3.** Schematic illustration of the physical model.

The density and viscosity ratios of liquid and gas phases are $\rho_l/\rho_g = 828$ and $\mu_l/\mu_g = 56$, respectively, with liquid and gas densities $\rho_l = 998.2$ kg/m$^3$ and $\rho_g = 1.205$ kg/m$^3$, respectively, and liquid and gas viscosities $\mu_l = 1.003 \times 10^{-3}$ Pa·s and $\mu_g = 17.9 \times 10^{-6}$ Pa·s, respectively; the liquid surface tension coefficient $\sigma = 0.072$ N/m.

In reality, when the droplets impact the film surface, instability occurs immediately at the moment of impact and grows over time, eventually leading to the formation of a crown spike. In order to generate this kind of perturbation in a numerical simulation to reflect the dynamics of a real droplet impact, an artificial perturbation, that is, a perturbation in velocity $U'$, is added to the initial droplet velocity $U_d$:

$$U' = \lambda U_d \exp(-(z-H)^2/D_d)\cos(n\theta), \tag{15}$$

where $\theta$ is the azimuthal angle, $n$ is the frequency, determining the number of the crown spikes, and $\lambda$ is the non-dimensional amplitude of perturbation. A high amplitude $\lambda$ enhances the rim instability, but does not affect the overall structure of the crown. In this work, a perturbation is imposed with an amplitude in the range of $0.2 \leq \lambda \leq 0.3$. The number of crown spikes is estimated using [37]:

$$n = 1.14\sqrt{We}. \tag{16}$$

Figures 4 and 5 show the numerical and experimental results of the interface evolution after droplet impact. The numerical conditions are: case A: $Re = 6875$, $We = 300$, $H = 0.094$, $\lambda = 0.2$, and $n = 16$; and case B: $Re = 6875$, $We = 300$, $H = 0.248$, $\lambda = 0.2$, and $n = 16$. From the simulation outcomes, the evolution of the crown and formation of the crown spike and satellite droplets are clearly captured, which is very consistent with the experimental results.

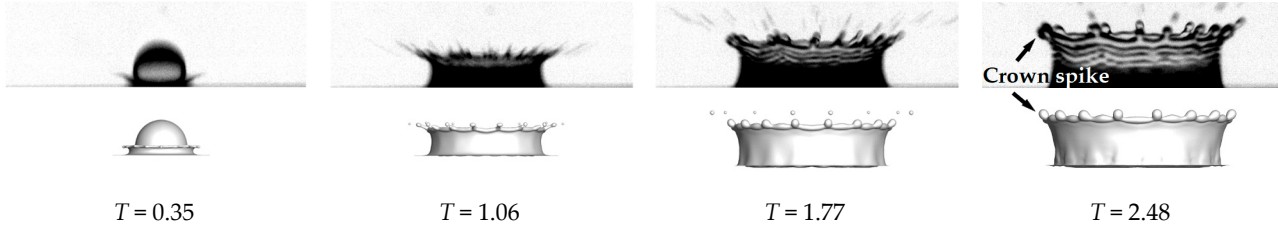

| $T = 0.35$ | $T = 1.06$ | $T = 1.77$ | $T = 2.48$ |

**Figure 4.** Comparison of crown profiles (case A). (**Top**) Experimental results. (**Bottom**) Simulation results.

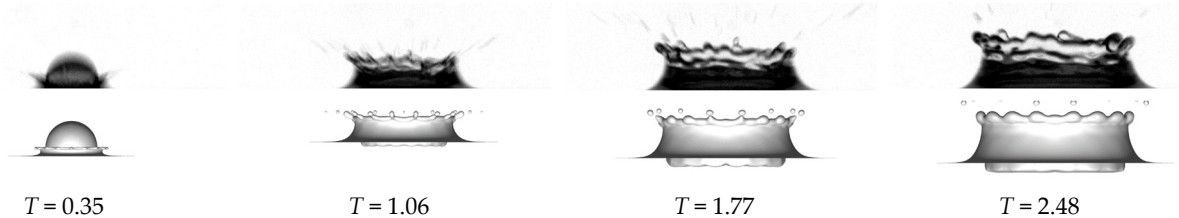

$T = 0.35$　　　　　$T = 1.06$　　　　　$T = 1.77$　　　　　$T = 2.48$

**Figure 5.** Comparison of crown profiles (case B). (**Top**) Experimental results. (**Bottom**) Simulation results.

In case A, the crown spike is relatively obvious, and satellite droplets are formed at the top of the crown spike due to surface tension. The crown spike is also relatively obvious, and it gradually lengthens over time. As the crown spike is further stretched and broken at the neck, the satellite droplets splash. The formation and splashing of satellite droplets occur several times during the evolution of the crown, and satellite droplets are generated more readily in early impact than in late impact. These results are considerably consistent with the experimental results.

In case B, the crown spike also appears at the early stage of impact, but then dissipates gradually. The reason for this is the comparatively long diffusion distance of the perturbation velocity in the thicker liquid film, which weakens the influence of the perturbation. In addition, the number of crown spikes in the experimental results is less than that of the numerical results at $T = 2.48$. Because it is difficult to ensure the uniform perturbation on crown edge in real experimental conditions, the crown spikes are not of uniform size and their evolution directions are not strictly perpendicular to the crown edge. It is possible that the two adjacent crown spikes gradually approach and eventually merge during the evolution of the crown.

To further validate our model, we carried out a quantitative comparison between the numerical and experimental results. As shown in Figures 6 and 7, the X-axis represents the dimensionless time $T$, and the Y-axis represents the dimensionless radius $R_c = r_c/D_d$ and dimensionless height $H_c = h_c/D_d$ of the crown. The dimensionless radius of the crown fulfills a square-root relationship with dimensionless time. This result is consistent with research by Ref. [4], and the numerical result is consistent with the experimental data, which further verifies the numerical method.

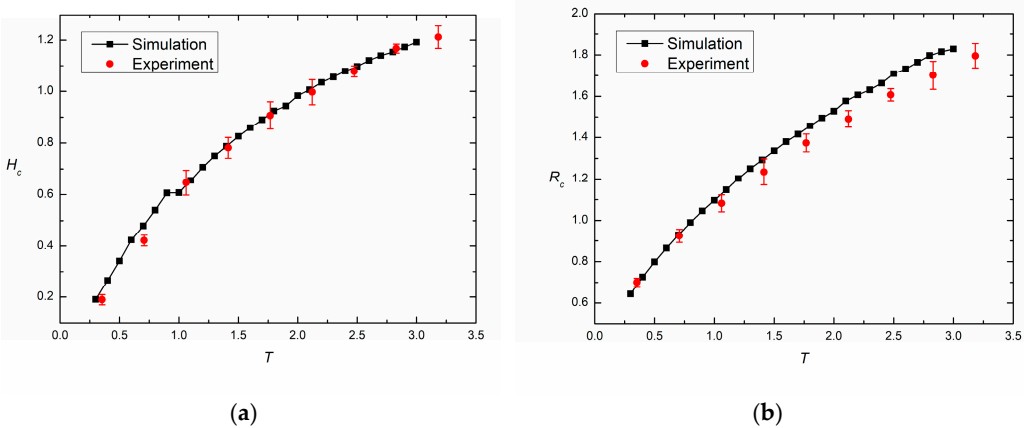

(**a**)　　　　　　　　　　　　　(**b**)

**Figure 6.** Comparison of crown heights (**a**) and radius (**b**) between the numerical and experimental results of Case A.

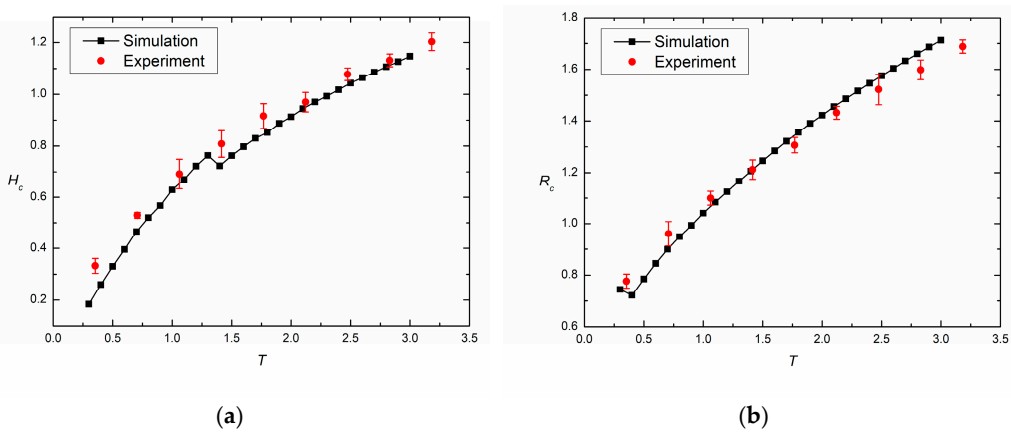

**Figure 7.** Comparison of crown heights (**a**) and radius (**b**) between the numerical and experimental results of Case B.

Figure 8 shows the numerical results and experimental results of the evolution of an oblique droplet impact on the liquid film (case C) with conditions: $Re = 7280$, $We = 177.5$, $H = 0.05$, $\theta = 30°$. The results indicate that the geometric features of the crown are not symmetrical in the X-direction, and the evolution of the crown is enhanced downstream, while inhibited on the upstream side. The numerical results are also quantitatively compared with the experimental data. As shown in Figure 9, the X-axis represents the dimensionless time $T$, and the Y-axis represents the dimensionless radius and dimensionless height of the crown on the downstream side. The numerical result is highly consistent with the experimental data.

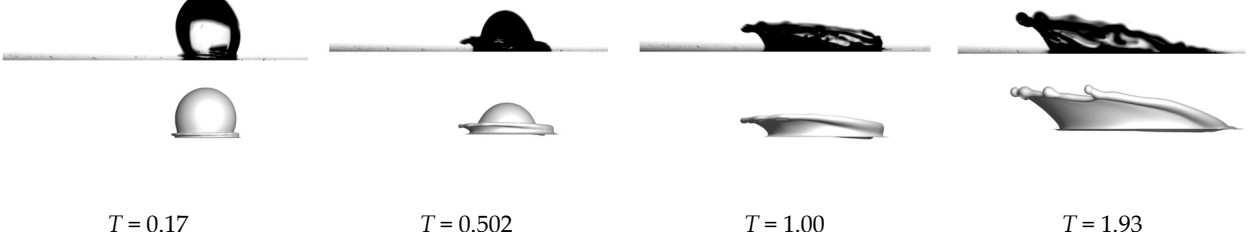

|  |  |  |  |
|---|---|---|---|
| $T = 0.17$ | $T = 0.502$ | $T = 1.00$ | $T = 1.93$ |

**Figure 8.** Comparison of the crown profiles (case C). (**Top**) Experimental results. (**Bottom**) Simulation results.

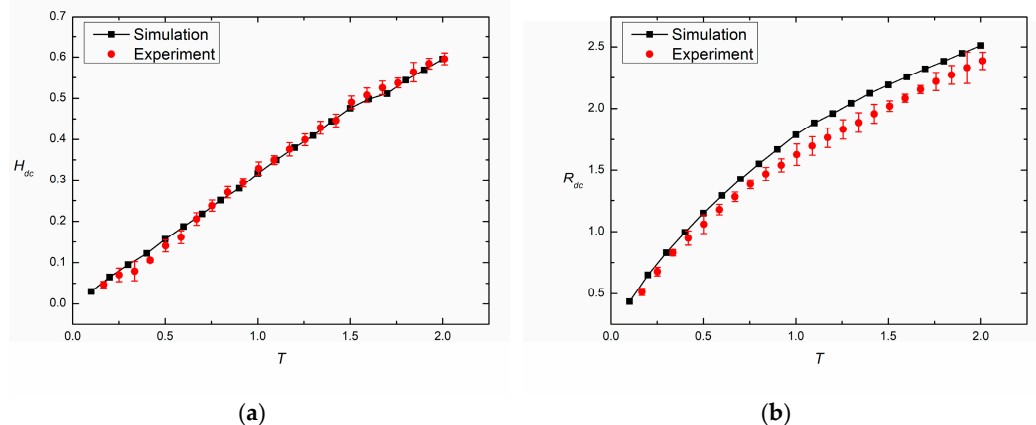

**Figure 9.** Comparison of downstream crown heights (**a**) and radius (**b**) between numerical and experimental results of Case C.

## 4. Results and Discussion

In the present chapter, the shape features of the crown, satellite droplet, attenuation law of momentum, and kinetic energy after an oblique drop impact on a liquid film are

discussed qualitatively and quantitatively. The main influencing factors of droplet impact are tangential velocity, film thickness, Reynolds number, and Weber number. The following reference parameters were used in the simulation: $\theta = 30°$, $h = 0.25$, $We = 250$, $Re = 597$, $D_d = 20$ μm, $U_d = 30$ m/s. The values are varied when the effects of specific parameters are discussed, while the other parameters keep the baseline values.

### 4.1. Effect of Tangential Velocity

Figure 10a,b show the evolution of the gas–liquid interface for a droplet impact at tangential velocities $U_{dt} = 6.96$ m/s and 15.0 m/s, corresponding, respectively, to the impact angles $\theta = 15°$ and $30°$, for a given normal impact velocity $U_{dn} = 25.98$ m/s. The annular crown lamella is ejected from the fusion location of the droplet and the film, gradually elongating outward with time. At the impact location, the hemispherical liquid surface gradually transforms into an inclined pit inside the annular crown. On the downstream side, the evolution of the crown is accelerated, and the instability of the crown edge is also increased. Near the upstream side, the situation is opposite. This asymmetry becomes more pronounced as the tangential velocity increases. At $T = 1.5$, the formation of satellite droplets is observed obviously, with $U_{dt} = 15.0$ m/s, but at $U_{dt} = 6.96$ m/s, only the crown spikes appear. This is partly due to the change in Weber number caused by the increase in impact velocity. Figure 11 displays the section of interface on the Y-mid with different tangential velocities. The downstream crown tends to bend downward at $U_{dt} = 25.98$ m/s.

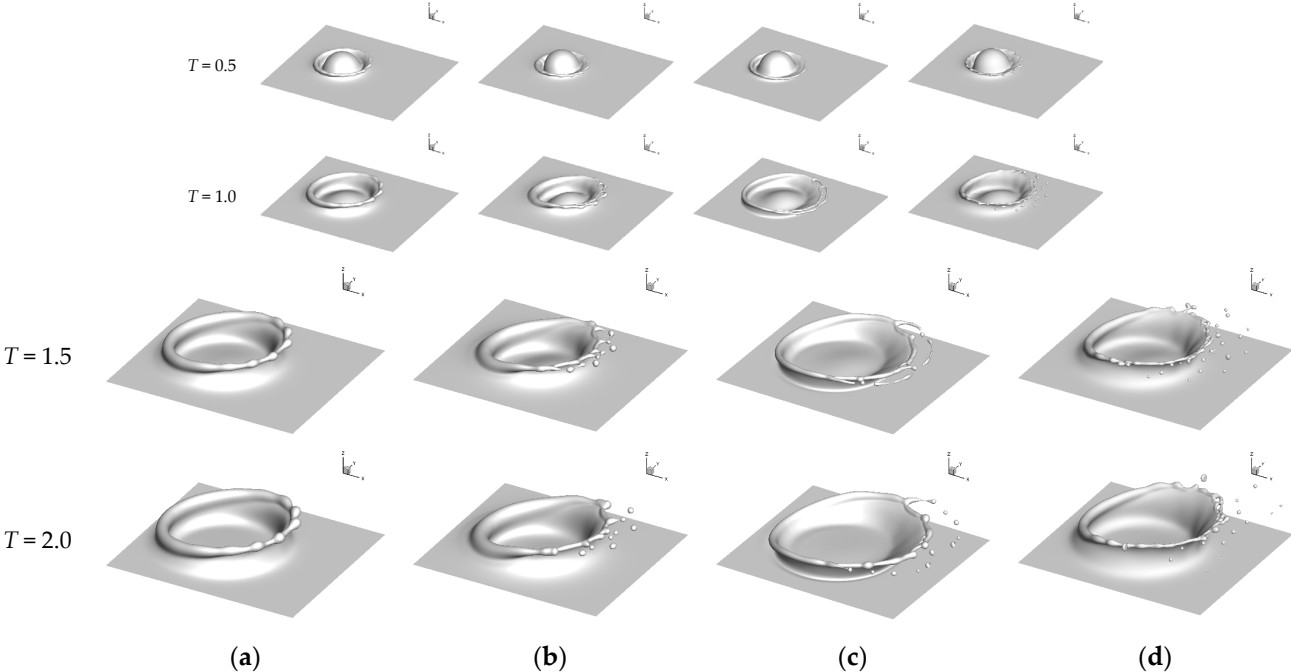

**Figure 10.** Flow patterns of the drop impacts on the thin film at different dimensionless times: (**a**) $\theta = 15°$, $H = 0.25$, $We = 200.64$, $Re = 535.43$. (**b**) $\theta = 30°$, $H = 0.25$, $We = 249.55$, $Re = 597.13$. (**c**) $\theta = 30°$, $H = 0.10$, $We = 249.55$, $Re = 597.13$. (**d**) $\theta = 30°$, $H = 0.25$, $We = 499.1$, $Re = 597.13$.

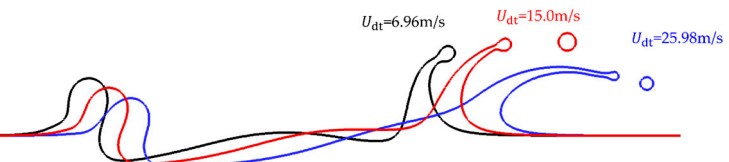

**Figure 11.** Section of the gas–liquid interfaces with different tangential velocities on the Y-mid plane at $T = 1.5$.

Figures 12 and 13 shows the time evolutions of dimensionless crown height and radius on the downstream and upstream sides. The crown height and radius increase with time evolution. For a given normal impact velocity, the height and radius of the upstream crown decrease with the increasing tangential velocity. On the downstream, the radius increases with the increase in tangential velocity. However, the height does not change significantly when $U_{dt}$ increases from 6.96 m/s to 15.0 m/s, but decreases obviously when $U_{dt}$ increases from 15.0 m/s to 25.98 m/s. This shows that tangential velocity inhibits crown evolution in the vertical direction.

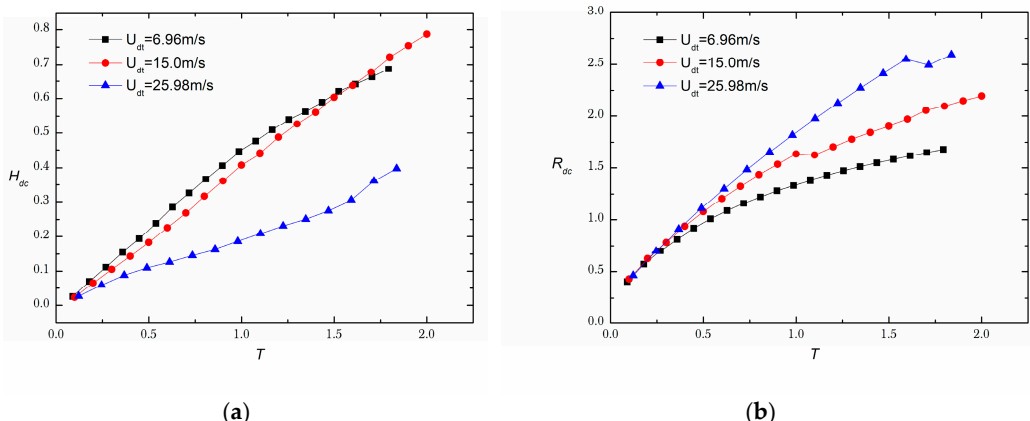

**Figure 12.** Time evolution of downstream crown height (**a**) and radius (**b**) at different tangential velocities.

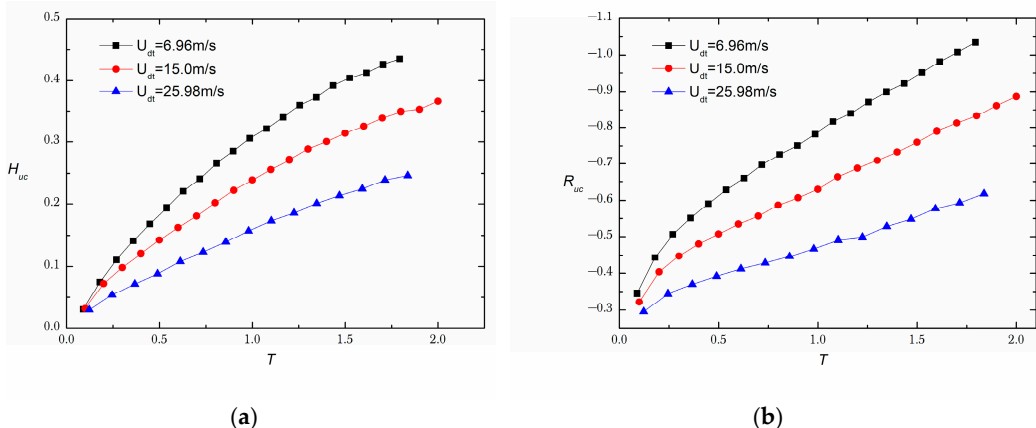

**Figure 13.** Time evolution of upstream crown height (**a**) and radius (**b**) at different tangential velocities.

The time evolution of the dimensionless momentum in the X-direction and dimensionless kinetic energy of the liquid phase in the whole computational domain are shown in Figure 14. The fastest momentum decay occurs in a time range of $0 \leq T \leq 0.1$, the initial stage of impact, for the most severe deformation of the gas–liquid interface. The momentum of the droplet is transferred to the gas between the droplet and film, as shown in Figure 15a. Then the momentum in the X-direction increases slightly in a short period of time due to momentum transfer from the Y-direction (normal to wall), because of the stagnation effect from the wall in the Y-direction. The momentum transfer from Y-direction decreases, and due to continuous momentum transfer from liquid to gas phase as shown in Figure 15b, the momentum in the X-direction decays with time. Similar to the change in momentum, the kinetic energy decays rapidly in the initial stage of impact, and decreases gradually with time.

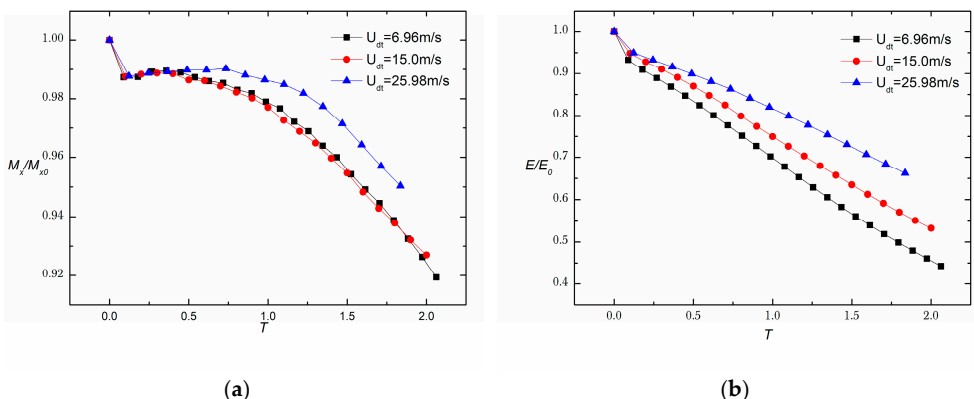

(**a**)　　　　　　　　　　　　　　(**b**)

**Figure 14.** Time evolution of the momentum in the X-direction (**a**) and kinetic energy (**b**) at different tangential velocities ($M_{x0}$ and $E_0$ are the momentum in the X-direction and kinetic energy at $T = 0$. The momentum we compare is the ratio of the non-dimensional momentum at the current moment to the initial non-dimensional momentum. The treatment of kinetic energy is the same as that of momentum).

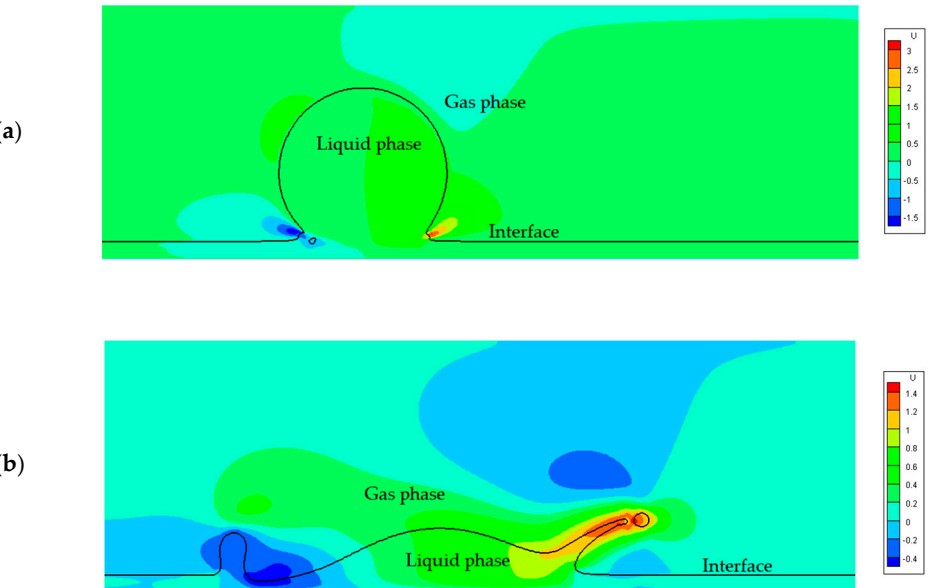

**Figure 15.** The distribution of the velocity component in the X-direction on the Y-mid plane at $T = 0.1$ (**a**) and $T = 1.0$ (**b**).

Varying the tangential velocity does not change the attenuation characteristics of the momentum in each time period, in general, but it does affect the decay of momentum in the X-direction. For a higher tangential velocity of 25.98 m/s, the momentum decay is obviously slower. However, when the normal impact velocity is fixed, changing the tangential velocity within the range of 6.96–15.0 m/s does not significantly affect the momentum attenuation in the X-direction during impact. For kinetic energy, because the Reynolds number and Weber number are no longer fixed due to the change in tangential velocity, the variation in kinetic energy is obvious, and the decrease in tangential velocity accelerates the decrease in kinetic energy.

### 4.2. Effect of Film Thickness

Figure 10b,c illustrate the time evolution of the crown height and the crown radius on the upstream and downstream sides at H = 0.25 and 0.1; the larger annular crowns are produced when the droplets hit a thinner water film. Figure 16 shows the section of interface on the Y−mid with different film thicknesses. Figure 17 shows that the crown

height decreases but the radius increases as film thickness decreases on the downstream side. The thinner the water film, the smaller the blocking effect on the crown evolution in the horizontal direction. This makes the angle between the crown and the horizontal surface smaller, resulting in a lower height but a larger radius. As shown in Figure 18, the variation of height and radius with film thickness on upstream are slightly different from the downstream side. The height of the upstream crown decreases when the film thickness increases. In addition, the crown height decreases after T = 1.4 on the upstream side at H = 0.1. This is because, in this time range, the tilt direction of the crown on the upstream side changes from a downstream tilt direction to an upstream tilt direction, which can be observed in Figure 16, and this phenomenon will not occur when other relevant parameters are changed.

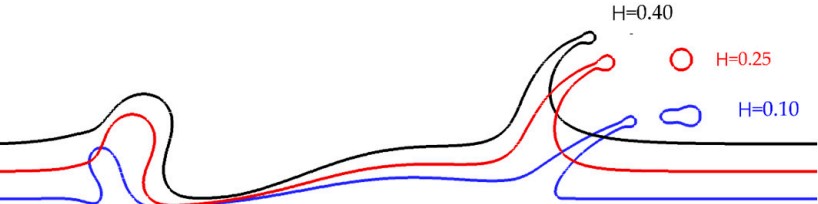

**Figure 16.** Section of the gas–liquid interfaces with different film thicknesses on the Y-mid plane at *T* = 1.5.

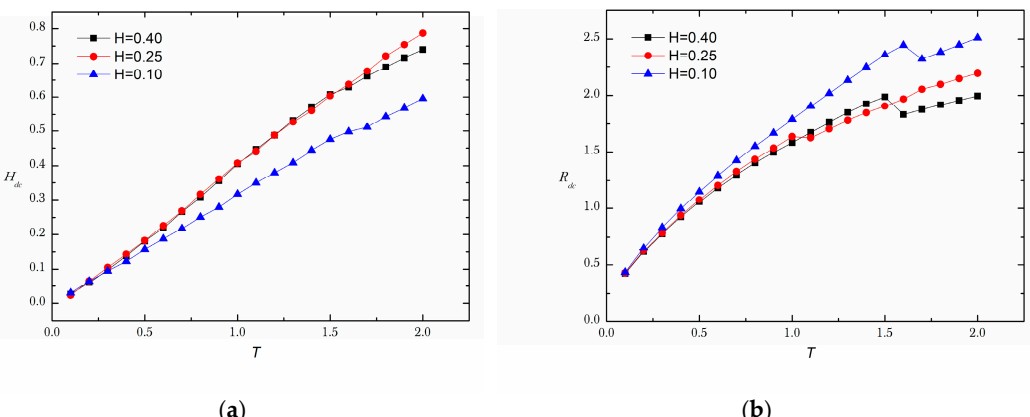

(**a**)                                    (**b**)

**Figure 17.** Time evolution of the downstream crown height (**a**) and radius (**b**) at different film thicknesses.

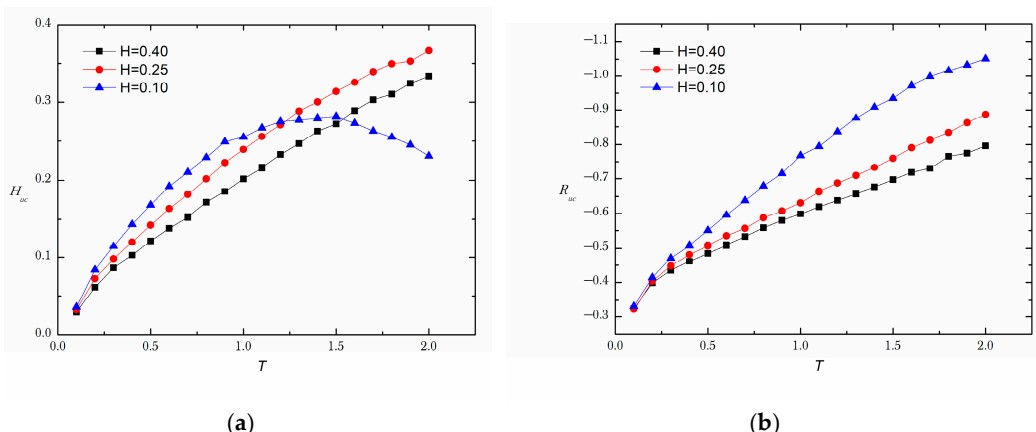

(**a**)                                    (**b**)

**Figure 18.** Time evolution of the upstream crown height (**a**) and radius (**b**) at different film thicknesses.

Figure 19 shows the decay of momentum in the X-direction and the kinetic energy with time of the liquid phase in the whole computational domain. Notably, an increase in liquid film

thickness reduces the momentum decay in X-direction and the kinetic energy. The reason for this phenomenon is that more kinetic energy is transferred to lower the film layers when the droplet impacts on a thicker film. It also explains why the crown evolves more slowly with a thicker film: less kinetic energy is transferred into the crown. In addition, with the further increase of the thickness, the influence of thickness on crown evolution, the momentum in the X-direction, and kinetic energy decrease is gradually weakened.

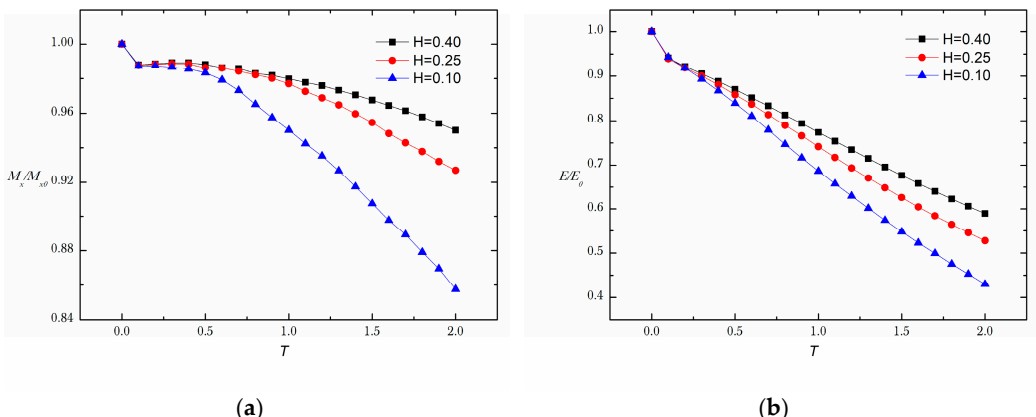

(a)                                              (b)

**Figure 19.** Time evolution of momentum in the X−direction (**a**) and kinetic energy (**b**) at different film thicknesses.

*4.3. Effect of Reynolds Number*

Figure 20 shows the section of crown geometric profiles on the Y-mid plane at Re = 597.13, 1333.59, and 1890.90. The influence of Reynolds numbers varieties is only reflected on the secondary droplets splash of the downstream crown and the evolution of the upstream crown. In addition, Figure 21 shows the time evolution of crown height and radius on the downstream side with different Reynolds number; there is only a slight increase in height and radius with the increasing Reynolds number. Figure 22 shows the evolution of the crown on the upstream side. The dimensionless height of the crown upstream slightly changes, while the radius obviously decreases with the increasing Reynolds number because the upstream crown tends to downstream due to a decrease of the viscous effect.

Figure 23a shows the time varying curve of the momentum in the X-direction. The decay of the momentum does not show a clear change rule with the increase of Reynolds number. Figure 23b shows the kinetic energy decrease curves at different Reynolds numbers, and that the decay is slower at a high Reynolds number. This is because the higher the Reynolds number is, the smaller the viscous force effect, and the less kinetic energy is transferred from the liquid phase to the gas phase because of the viscous effect.

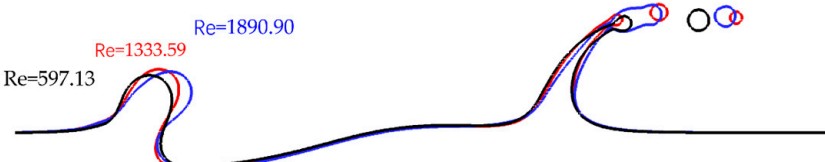

**Figure 20.** Section of the gas–liquid interfaces with different Reynolds numbers on the Y-mid plane at *T* = 1.5.

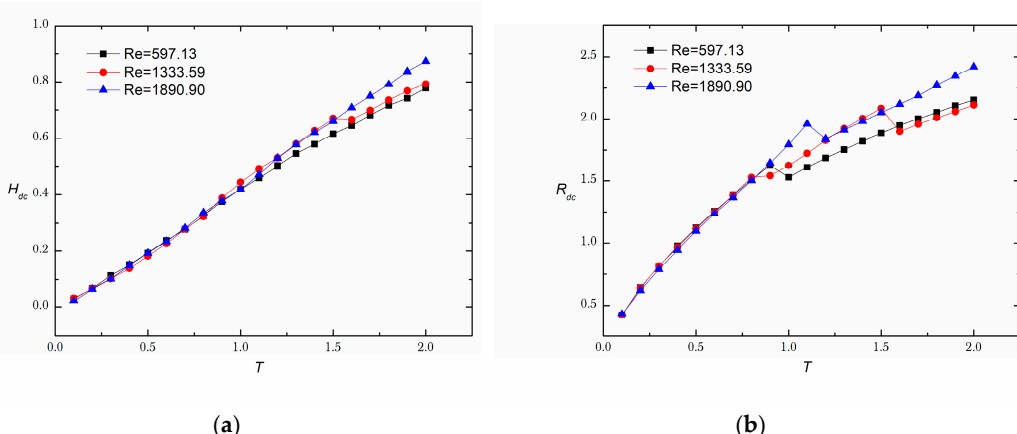

**Figure 21.** Time evolution of downstream crown height (**a**) and radius (**b**) at different Reynolds numbers.

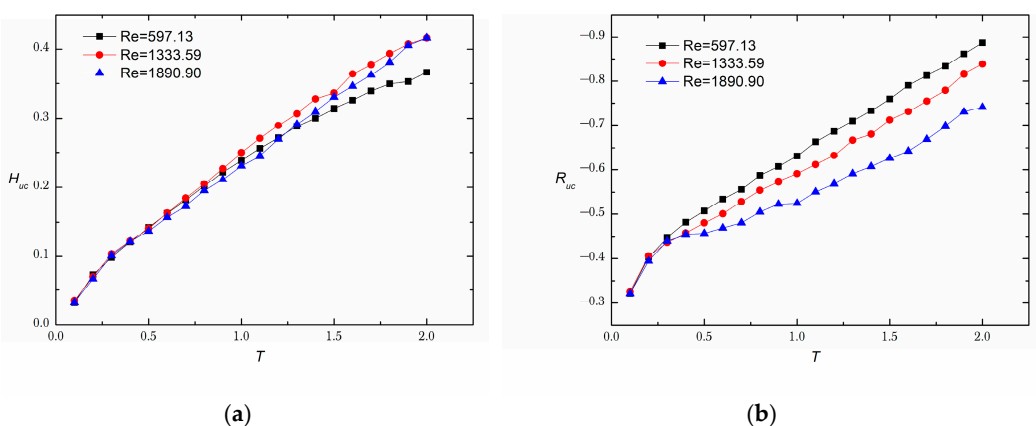

**Figure 22.** Time evolution of upstream crown height (**a**) and radius (**b**) at different Reynolds numbers.

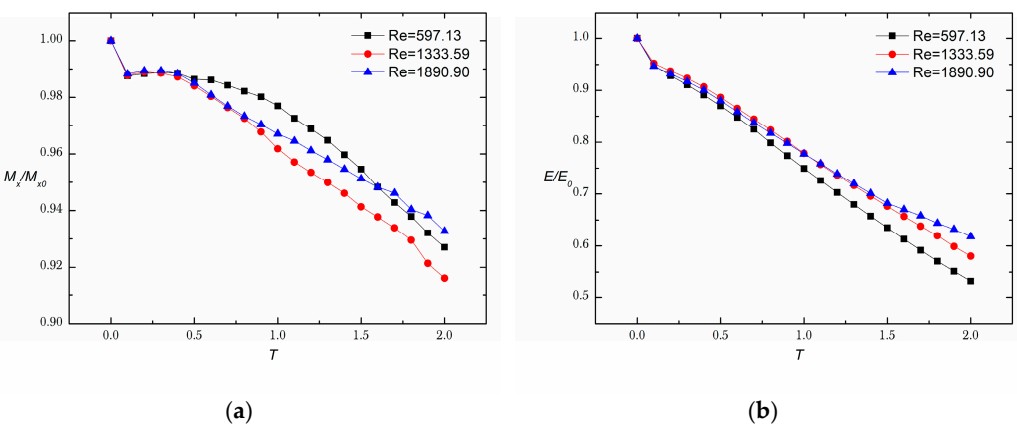

**Figure 23.** Time evolution of the momentum in the X−direction (**a**) and kinetic energy (**b**) at different Reynolds numbers.

### 4.4. Effect of Weber Number

The time evolution of the interface of the oblique droplet with different Weber numbers (249.55 and 499.1) is shown in Figure 10b,d. A large number of satellite water droplets are observed in the droplet impact at We = 499.1. A higher Weber number means that the crown elongates faster, and the crown rim breaks more easily, generating satellite droplets. In addition, from the gas–liquid interfaces (Figure 24) and the time evolution of the height of the upstream (Figure 25) and downstream crowns (Figure 26), it is clear that the higher the Weber number, the lower the surface tension, which leads to an increase in the height of the upstream and downstream crowns.

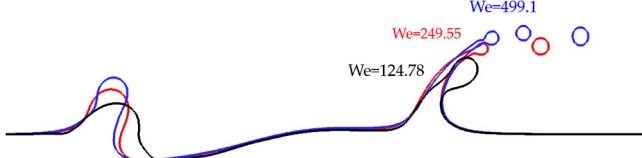

**Figure 24.** Section of the gas–liquid interfaces with different Weber numbers on the Y-mid plane at *T* = 1.5.

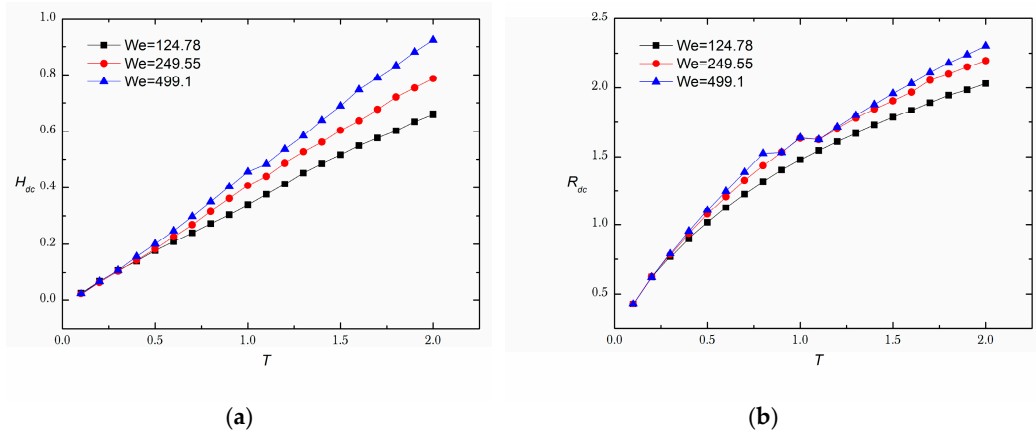

(**a**)　　　　　　　　　　　　　　　　(**b**)

**Figure 25.** Time evolution of downstream crown height (**a**) and radius (**b**) at different Weber numbers.

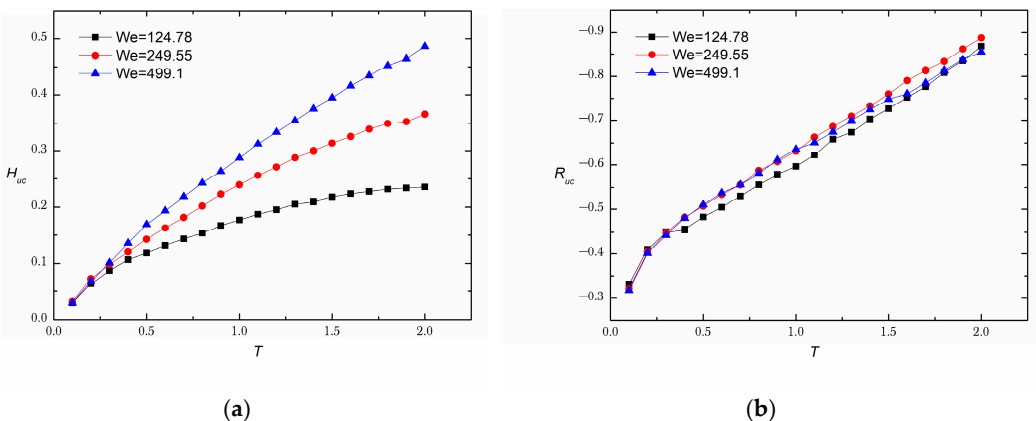

(**a**)　　　　　　　　　　　　　　　　(**b**)

**Figure 26.** Time evolution of upstream crown height (**a**) and radius (**b**) at different Weber numbers.

High Weber numbers promotes the decay of momentum in X-direction, as shown in the Figure 27a. The curves of kinetic energy with time are completely different from that of the momentum in the X-direction for different Weber numbers. Although the kinetic energy is attenuated more at low Weber numbers, the difference in value is very slight. Moreover, before *T* = 1.5, the variation of momentum decay with different Weber numbers is not distinct. Overall, the kinetic energy decay curves of different Weber numbers are basically coincident, which confirms that the attenuation of kinetic energy is not sensitive to the Weber number.

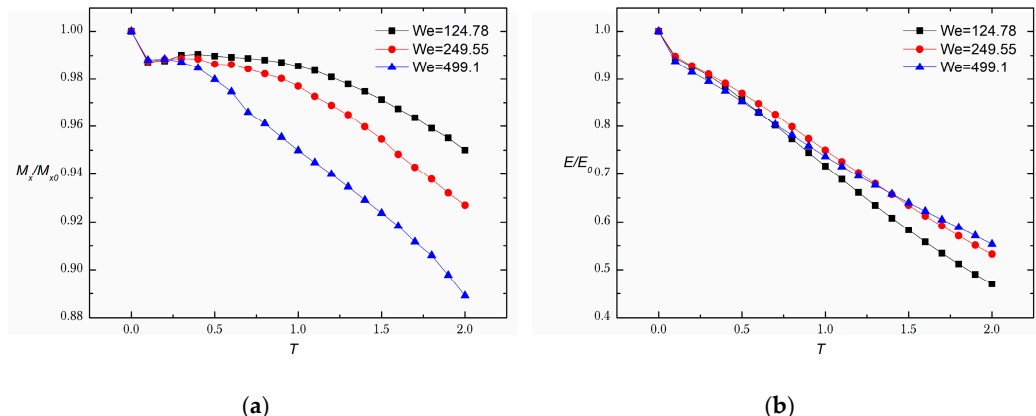

**Figure 27.** Time evolution of the momentum in X−direction (**a**) and kinetic energy (**b**) at different Weber numbers.

*4.5. Decay Rate of the Momentum in X-Direction*

From the above discussion, we can see that momentum in the X-direction is generally decreasing with the different rates. To further study the characteristics of momentum decay, the curves of momentum decay rate $R_m$, representing the temporal derivative of momentum, with time, are shown in Figure 28. It can be seen that the momentum decay rate suddenly becomes positive due to the momentum transfer from the Y-direction and then gradually increases. Compared with the change of tangential velocity (Figure 28a) and Reynolds number (Figure 28c), the change of film thickness (Figure 28b) and Weber number (Figure 28d) present obvious regularity to the decay rate. The thicker water film can preserve more impact momentum, so when film thickness increases, the momentum decay rate tends to decrease. At $T$ = 1.8, when the film thickness changes from 0.1 to 0.25, the momentum decay rate decreases by 44.9%, and it decreases by 62.2% when film thickness increases to 0.4. In addition, the momentum decay rate increases with the Weber number, because the crown evolves faster when the surface tension is smaller, resulting in more momentum transfer to the gas phase. At $T$ = 1.8, the momentum decay rate increases by 26.8% when the Weber number increases from 124.78 to 249.55, and increases by 63.0% when the Weber number is 499.1. As shown in Figures 16 and 24, thicker water film and stronger surface tension force slow down the evolution of the crown. Correspondingly, the momentum decays with the smaller rate.

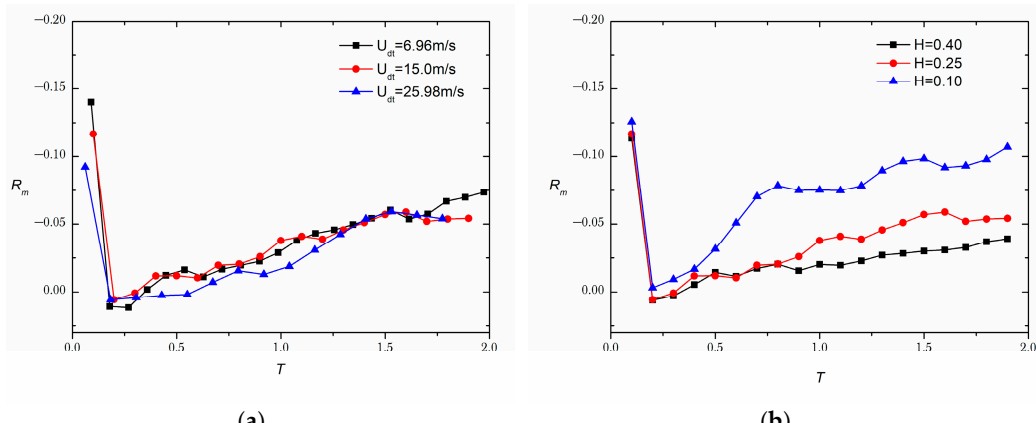

**Figure 28.** *Cont.*

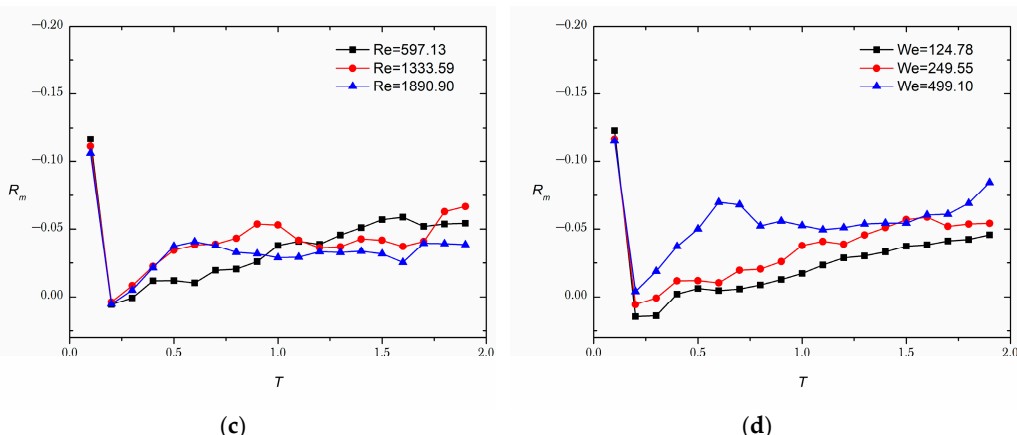

**(c)**                                                  **(d)**

**Figure 28.** Decay rate of the momentum in the X-direction with different parameters. (**a**) tangential velocity; (**b**) film thickness; (**c**) Reynolds number; (**d**) Weber number.

## 5. Conclusions

In the present study, the oblique droplet impact dynamics on a liquid film were numerically investigated. The level-set method was employed to capture the gas–liquid interface, and Navier–Stokes equations were solved on collocated grids. The accuracy of the numerical method was verified by its good consistency with the experimental results. In our numerical investigation, using conventional environmental parameters for aircraft icing, we simulated microscale water droplets (20–100 μm in diameter), with an initial impact velocity of 10–30 m/s, impact angle of 15–45°, and relative film thickness of 0.1–0.4. The evolution of the crown features on the upstream and downstream sides after impact and under different conditions were discussed. The variation in momentum in the X-direction and kinetic energy of liquid phase during impact were also investigated. The concluding remarks are as follows:

(1) Increasing the tangential velocity and Weber number promote crown evolution, while the increasing film thickness reduces the crown height and radius. Moreover, the effect of film on crown becomes more obvious when the film is thinner. The change of Reynolds number has very limited effect on the crown.

(2) The decay of momentum in the X-direction of the oblique impact of a droplet was divided into three stages: short but rapid decrease, slight increase, and continuous decrease. However, the kinetic energy always decreases (at a gradually decay rate).

(3) The decay rate of momentum in the X-direction initially decreases and then increases with time. Decreasing the thickness of the water film and increasing the Weber number accelerates the evolution of the crown and therefore, the momentum in the X-direction decays with a larger rate.

**Author Contributions:** Conceptualization, Y.C. and J.W.; methodology, Y.C.; software, Y.C.; validation, Y.C. and J.W.; formal analysis, Y.C.; investigation, Y.C.; resources, C.Z.; data curation, Y.C.; writing—original draft preparation, Y.C.; writing—review and editing, Y.C.; visualization, Y.C.; supervision, C.Z.; project administration, C.Z.; funding acquisition, C.Z. All authors have read and agreed to the published version of the manuscript.

**Funding:** This research was funded by the Natural Science Foundation of China (NSFC Grant No. 11832012).

**Institutional Review Board Statement:** Not applicable.

**Informed Consent Statement:** Not applicable.

**Data Availability Statement:** The data that support the findings of this study are available from the corresponding author upon reasonable request.

**Acknowledgments:** We would like to express our thanks to the editors of Aerospace and the reviewers for their work in processing this article.

**Conflicts of Interest:** The authors declare no conflict of interest.

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
