# Peer review of "Numerical Simulation of Microscale Oblique Droplet Impact on Liquid Film"

_aerospace, doi:10.3390/aerospace10020119_

Round 1

Reviewer 1 Report

The manuscript reports an investigation of oblique droplet impact on thin wall-films. Level set method based computational investigation is adopted in this study. Very few experiments are conducted at reference conditions to validate the computational approach. The main parameters investigated are tangential velocity, film thickness, Reynolds number and Weber number. The effect of these parameters on the time evolution of crown geometrical parameters (downstream and upstream heights and radii of the crown) as well as that of the momentum in x-direction and kinetic energy are reported and discussed.

The topic of this study is relevant and could be of interest to the readers However, the novelty of the study, compared to other reported in the literature, is not clear. As mentioned in the manuscript, if the momentum and kinetic energy evolutions are the novel contributions of this study, these aspects are not discussed in detail in the manuscript. Furthermore, there are numerous ambiguities in the scientific discussion (listed as 'Major Comments' below) as well as language-related mistakes (listed as 'Minor Comments' below). The authors should address these carefully in a revised manuscript.

Major Comments:

1. Lines 51-52: How are 'thin' and 'thick' films defined? In other words, what parameter is used to characterize the transition from thin film to thick film? Please mention this in the revised manuscript.

2. The details of the droplet size and velocity measurements are missing in the experimental methodology. Furthermore, the surface roughness and wetting characteristics of the smooth aluminum plate is not mentioned. These are also important factors since surface wetting governs the uniformity of the thickness of the film formed on the surface. How easy was it to form a thin film of water uniformly on the aluminum surface? Please discuss these aspects in the revised manuscript.

3. Line 196: This statement gives the impression that there are two dimensionless times when the droplet contacts with the film surface. Shouldn't this be rather T = tUd/Dd = 0? Re-phrase the statement accordingly in the revised manuscript.

4. Line 205: What does X = -1.0 imply? Please clarify in the revised manuscript.

5. Line 207: Does D mean Dd here? Please clarify and be consistent with the notations used in the manuscript.

6. Line 223: Isn't \lambda*Ud the amplitude of perturbation? Please correct in the revised manuscript.

7. Eq. 15: The number of crown spikes/fingers is artificially introduced as an input in the simulations. The input is, in turn, based on the impact condition and obtained from an empirical relation reported in the literature. How is this justified? One would expect n to be an output from the simulations/experiments. Please clarify this in the revised manuscript.

8. Eq. 15: This relation is valid for droplet impact on dry solid surface. What is the justification for using this for the current problem of droplet impact on wall-film?

9. What is the difference between crown spike and finger? Please clearly define these structures in one of the pictures in Fig. 3.

10. Line 249: The merging of fingers is not captured well in the simulations. What could be the reason for this? Please discuss this in the revised manuscript.

11. Fig. 9c: Detachment of rim as a continuous thread-like structure is seen here at T = 1.0 and beyond. Is this physically realistic or is this due to numerical instability in the simulations? Please clarify the reason for the observed detachment of rim.

12. Sec. 4.1: Phrases such as 'the droplet melts into a liquid film' and 'diffusion speed of the crown' are used without much care. These phrases mean something different in the scientific terminology. For example, there is no 'melting' (phase change) occurring in this study. Please rephrase such terms in the revised manuscript. 

13. Line 312: Figure 11 does not show upstream parameters of the crown geometry. Please correct this statement accordingly in the revised manuscript.

14. Lines 323-324: Why does the momentum in the x-direction increase after the initial stage? The reason given in the manuscript ('partly transfer of momentum from other directions') is not clear. Which other directions are being referred to here? And, what is the mechanism by which it gets transferred? Please clarify this in the revised manuscript.

15. Lines 325-327: The momentum decreases with increase in time after the initial stage and the short increase thereafter. What are the reasons for this continuous decrease in momentum in the x-direction?

16. Lines 337-338: This is opposite to the trend seen in Fig. 13b. Please check and clarify/correct the statement in the revised manuscript.

17. Lines 342-34: These statements which discuss the trend of crown geometry with film thickness are directly opposite to what is observed in Fig. 14, where crown height increases and crown radius decreases as film thickness increases.

18. Why does Fig. 17 show the parameters evaluated for the whole computational domain whereas Fig. 13 shows them for only the liquid phase? Similarly, it is not clear how the parameters are evaluated in Fig. 25 (are they for the liquid phase or for the whole computational domain?).

19. Lines 362-364: How is a reduction in the 'decay rate of momentum in x-direction' related to a slow evolution of crown? Please clarify in the revised manuscript.

20. Lines 366-367: How does energy in the crown decrease with decrease in film thickness? The statement before this indicates otherwise. Please clarify in the revised manuscript.

21. Fig. 20b: Why is Ruc more at same time instant for a lower Re?

22. Phrases such as 'decay loss rate' and 'decay rate' are used interchangeably. It is not clear what 'decay loss rate' is! Please be consistent and clear with the terminologies used. 

23. How are different Weber numbers realized maintaining the same Reynolds number?

24. Lines 417-419: Do the authors mean 'momentum decay' or 'kinetic energy decrease' here?

Minor Comments:

1. Line 26: Replace 'inkjet painting' with 'inkjet printing'.

2. Eq. 1: Include the non-dimensional wall-film thickness also in the list of dimensionless parameters.

3. Line 41: Replace "Ud and Dd are the droplet diameter and impact velocity" with "Dd and Ud are the droplet diameter and impact velocity". 

4. Line 80: Replace 'disposition' with 'deposition'.

5. Line 99: Replace 'existed' with 'exhibited'.

6. Line 101: 'the regulation is contrary' can be replaced with 'the trend is opposite'.

7. Lines 107-108: What do the authors mean by this? Do the authors mean that the laws of conservation of momentum and energy have not been used in the context of the previous studies? Or, do the authors mean that the changes in momentum and kinetic energy of the film and droplet are not explicitly studied? Please clarify in the revised manuscript.

8. Line 132: Replace 'defined' with 'is defined'.

9. Eq. 7: What is the difference between second and third terms within the bracket in H(\phi) for |\phi| <= \epsilon?

10. Lines 158-159: Replace 'scheme of the advection terms as' with 'scheme as the advection terms of'.

11. Line 158: Replace 'solved' with 'is solved'.

12. Line 185: Do the authors mean that the LED light source was used in back-lighting/shadowgraph mode with ground glass as light diffuser? Please clarify in the revised manuscript.

13. Line 221: Replace 'are added' with 'is added'.

14. Line 261: Replace '60' with '60 deg.'.

15. Lines 315-316: The terms 'horizontal impact velocity' and 'tangential velocity' are used. Please stick to one of these terminologies uniformly throughout the manuscript.

Author Response

Dear Editors and Reviewers,

Thank you very much for your review and comments concerning our manuscript entitled “Numerical Simulation of Microscale Oblique Droplet Impact on Liquid Film (ID: aerospace-2032750)”. Especially, we sincerely appreciate you for giving us a chance to improve our manuscript, as well as the important guiding to our work. Your comments are all valuable and very helpful for improving our paper. We have revised our manuscript carefully in accordance with your comments. The changes are marked in red in the revised manuscript. Please find our detailed responses to your comments in the attached PDF.

Reviewer 2 Report

This is a numerical study on the oblique droplet impact of a droplet on a liquid film. Experimental validation is provided. The topic is still very relevant despite of the large number of studies already reported in the literature. The paper is well written and well organized. The results are well presented and discussed. However, and although I understand the arguments of authors regarding a deeper analysis on energy and momentum, I actually do not see that in the results, which present a rather classical approach, on the effect of the Weber and Reynolds number (which is an obvious and correct approach). There is some qualitative discussion, but I think that the results should stress the original analysis in a clearer manner, and for that, for instance, comparison with the vast literature that exists on this topic can be addressed.

Apart from this, there are a few more specific points that I would like to see addressed in a revised version of the manuscript:

-         Page 4, Governing equations: why are you addressing a multiphase flow? And how are the different phases handled?

-         How many impacts are taken for the experimental measurements of Hc and Rc? The diameters, velocities, etc are taken by image analysis, right? There is an uncertainty in this analysis that must be provided. For instance, there is a detailed and relatively recent (2018-2019) work of Roisman and co-authors on droplet impact and crown formation that may be revisited. It depends on the thickness of the film but they vary it in the different publications.

-         It would be better to provide the range of size and velocity of the droplets in the begin of the paper, e.g. in the validation model or in the numerical procedure. It is in the conclusions, but it is very late in the paper.

-         Please add error bars in the experimental data.

Author Response

Dear Editors and Reviewers,

Thank you very much for your review and comments concerning our manuscript entitled “Numerical Simulation of Microscale Oblique Droplet Impact on Liquid Film (ID: aerospace-2032750)”. Especially, we sincerely appreciate you for giving us a chance to improve our manuscript, as well as the important guiding to our work. Your comments are all valuable and very helpful for improving our paper. We have revised our manuscript carefully in accordance with your comments. The changes are marked in red in the revised version. Please find our detailed responses to your comments in the attached PDF.

Round 2

Reviewer 1 Report

The manuscript could be accepted for publication.

Reviewer 2 Report

Authors revised the paper according to reviewers' comments. For me it is OK to be published